# Preliminary Evidence on Pulmonary Function after Asymptomatic and Mild COVID-19 in Children

**DOI:** 10.3390/children9070952

**Published:** 2022-06-25

**Authors:** Costanza Di Chiara, Silvia Carraro, Stefania Zanconato, Sandra Cozzani, Eugenio Baraldi, Carlo Giaquinto, Valentina Agnese Ferraro, Daniele Donà

**Affiliations:** 1Division of Pediatric Infectious Diseases, Department of Women’s and Children’s Health, University of Padua, 35128 Padua, Italy; sandracozzani@gmail.com (S.C.); carlo.giaquinto@unipd.it (C.G.); daniele.dona@unipd.it (D.D.); 2Unit of Pediatric Allergy and Respiratory Medicine, Department of Women’s and Children’s Health, University of Padua, 35128 Padua, Italy; silvia.carraro@unipd.it (S.C.); stefania.zanconato@aopd.veneto.it (S.Z.); valentinaagnese.ferraro@unipd.it (V.A.F.); 3Neonatal Intensive Care Unit, Department of Women’s and Children’s Health, University of Padua, 35128 Padua, Italy; eugenio.baraldi@unipd.it

**Keywords:** COVID-19, long-covid, pulmonary function, spirometry, children

## Abstract

Background: While it has been described that adults can develop long-lasting deterioration in pulmonary function (PF) after coronavirus disease 19 (COVID-19), regardless of disease severity, data on the long-term pneumological impact of SARS-CoV-2 infection in children are lacking. Methods: Performing a single-center, prospective, observational study on children aged 6–18 years with a previous diagnosis of asymptomatic/mild COVID-19, we evaluated the long-term impact of mild severe acute respiratory syndrome coronavirus 2 (SARS-CoV-2) infection in children. Results: A total of 61 subjects underwent spirometry after a mean time of 10 ± 4 months from asymptomatic or mild infection. None of the children reported any respiratory symptoms, needed any inhaled therapy, or had abnormal lung function. Conclusions: In our study, we observed that children and adolescents did not develop chronic respiratory symptoms and did not present lung function impairment after asymptomatic or mild SARS-CoV-2 infection.

## 1. Introduction

Children can be infected by SARS-CoV-2 but are largely spared from a severe respiratory illness compared with adults [1]. Nevertheless, it has been proved that about 6% of children may report respiratory symptoms one month after COVID-19 regardless of the infection’s severity [2]. While it has been described that an adult can develop long-lasting deterioration in pulmonary function (PF) after COVID-19, preliminary data in the pediatric population show that children have normal lung function after recovery from mild disease [3,4]. However, reports on possible long-term PF sequelae in children and adolescents are lacking, and the few available data refer to a follow-up limited to the first few months after infection [3].

To gain a greater understanding of the long-term pneumological impact of SARS-CoV-2 infection in children, we herein describe the PF after SARS-CoV-2 infection in a pediatric cohort mostly presenting asymptomatic or mild disease, recruited at the Department of Women’s and Children’s Health (W&CHD) of Padua University Hospital (Italy).

## 2. Materials and Methods

We conduct a single-center, prospective, observational study in children aged 6–18 years who experienced COVID-19 in their family cluster and attended the COVID-19 Family Cluster Follow-up Clinic set up at the W&CHD. From August to November 2021, subjects were consecutively enrolled if they had a record of virological or serological positivity for SARS-CoV-2. Children younger than 6 years were not included because they are not able to perform technically acceptable and repeatable spirometry. Exclusion criteria were: (a) multisystemic inflammatory syndrome (MIS-C); and (b) a pre-existing chronic respiratory disease. At enrolment, a pediatrician and/or an infectious diseases specialist collected data on demographic parameters, past medical history, clinical features (including date of symptoms onset, date of close household contacts, date of the positive nasopharyngeal swab), and vaccinal status (including the SARS-CoV-2 vaccine). A blood sample was collected from all cases for diagnosis of previous SARS-CoV-2 infection through the evaluation of the immunological response to SARS-CoV-2. For each confirmed SARS-CoV-2 infection, a baseline date was defined: (1) considering the first date between the onset of symptoms or first positive SARS-CoV-2 molecular assay; and (2) for asymptomatic cases with negative/undetermined nasal-pharyngeal swab, the date was established within the family outbreak, coinciding with the date of symptoms onset in the family cluster. The severity of COVID-19 was scored following the World Health Organization (WHO) classification [5]. At least two months after infection, the recruited children were evaluated by a pediatric pulmonologist and their personal history of respiratory symptoms (i.e., chest tightness, wheezing, cough, and exercise-induced respiratory symptoms) since SARS-CoV2 infection was collected. Then, the children performed spirometry.

Spirometry was performed with a 10-L bell spirometer (Biomedin, Padua, Italia). The maneuver with the largest sum of forced vital capacity (FVC) and forced expiratory volume in the first second (FEV1) was considered. A bronchodilator reversibility test was carried out in case of airway obstruction. All spirometric values were analyzed using Z-score according to the reference values of the Global Lung Function Initiative (GLI) powered by the European Respiratory Society [6,7].

Data are summarized as mean (SD) or median (IQR) for quantitative variables, and as counts and percentages for categorical variables. Normality was checked with the Shapiro–Wilk test. Quantitative variables were compared across groups with t-test for independent variables. Pearson’s R coefficient was used for correlations. Statistical significance was set at *p*-value < 0.05. All statistical analyses were performed using SPSS 23.0 (IBM Corporation, Armonk, NY, USA).

The study was approved by the local Ethics Committee (Protocol No. 0070714), and all parents gave their written informed consent to their children’s participation in the study.

## 3. Results

From August to November 2021, we evaluated 66 children with confirmed SARS-CoV-2 infection. Five patients with pre-existing asthma were excluded. A total of 61 subjects (32 [52.5%] females) with a mean age of 10.9 ± 2.9 years were studied after a mean time of 10 ± 4 months from infection. None of the children had received any dose of COVID-19 vaccine before infection or before pneumological evaluation. During the acute phase of infection 24 (39.3%) children were asymptomatic and 37 (60.7%) were mildly symptomatic. None developed pneumonia, none received any anti-COVID-19 treatments.

Demographic characteristics, infection course, and outcome are summarized in Table 1.

At the pediatric pulmonologist’s evaluation, none of the children reported any respiratory symptoms or needed any inhaled therapy (e.g., bronchodilator, steroids), both at rest and after physical activities (Table 1).

The spirometric parameters evaluated (FEV1, FVC, FEV1/FVC, and FEF25/75) were normal in all the recruited children. Moreover, four children underwent a bronchodilator reversibility test, but none of them had a significant increase in FEV1. Spirometric values are summarized in Table 2.

FEV1: Forced expiratory volume in the first second; FVC: Forced vital capacity; FEF 25/75%: Forced expiratory flow at 25–75% of FVC.

No correlation was found between lung function parameters and the number of months since infection (FEV1 R: −0.165, *p* = 0.204, FVC R: −0.045, *p* = 0.732, FEV1/FVC R: −0.225, *p* = 0.081, FEF25/75 R: −0.209, *p* = 0.106).

Moreover, no difference was found in lung function (FEV1 *p* = 0.273, FVC *p* = 0.38, FEV1/FVC *p* = 0.702, FEF25-75 *p* = 0.356) comparing children from 6 to 10 years of age (*n* = 33, 54.1%) with children > 10 years old (*n* = 28, 45.9%).

## 4. Discussion

In our study, one of the largest Italian pediatric cohorts, we evaluated the long-term impact of mild SARS-CoV-2 infection in children, finding no effect on respiratory symptoms and lung function. Long-term loss of PF, due to an abnormal inflammatory response, was previously observed in children after respiratory viral infections, especially given by RSV and Rhinovirus [8,9,10]. In addition, preliminary results on long-covid in children showed that more than 40% presented at least one symptom >60 days after acute infection, and some of them developed respiratory symptoms after COVID-19 [2]. Therefore, we investigated PF in children after COVID-19, in order to assess whether asymptomatic or mild SARS-CoV-2 infection could sub-clinically impair respiratory parameters. In our population of children with a history of asymptomatic or mildly symptomatic SARS-CoV-2 infection, we found, indeed, no persistent respiratory symptoms (either at rest or on exertion) and normal lung function. Our results are in keeping with two previous pediatric studies [3,4] that found no effect on lung function in the first 6 months after asymptomatic or mild SARS-CoV-2 infection. On the other hand, focusing on a sub-group of children affected by a more severe SARS-CoV-2 infection, Knoke et al. observed a significant reduction in some spirometric parameters (i.e., FVC and MEF75) and impaired DLCO [4]. Our study has several limitations. First, including only children with mild or asymptomatic SARS-CoV-2 infection, we cannot provide any information on the possible detrimental effect of more severe cases (which are indeed very rare during childhood). Secondly, we performed only spirometry and did not examine the pulmonary gas exchange capacity.

## 5. Conclusions

In conclusion, we observed that children and adolescents did not develop chronic respiratory symptoms and did not present lung function impairment after asymptomatic or mild SARS-CoV-2 infection. Further studies are needed to confirm our findings and to investigate the possible long-term effects of COVID-19 on lung function in children with moderate/severe infection.

## Figures and Tables

**Table 1 children-09-00952-t001:** Demographic and clinical features of the 61 enrolled subjects.

Characteristics	Results
Number of patients	61
Age (years, mean, SD)	10.9 ± 2.9
Sex (male/female)	29/32
COVID-19 vaccination	0 (0%)
Practice sport regularly	40 (65.6%)
COVID-19 WHO classification	
Asymptomatic	24 (39.3%)
Mild	37 (60.7%)
Moderate	0 (0%)
Severe	0 (0%)
Anti-COVID-19 therapies	(0%)
Time from baseline to spirometry (months, mean, SD)	10 ± 4
Respiratory symptoms after COVID-19	
Symptoms at rest	0 (0%)
Exercise-induced respiratory symptoms	0 (0%)
Comorbidities (Other than respiratory chronic disease)	22 (36.1%)
Pre-school wheezing	3 (13.6%)
Gastrointestinal disease	2 (9.1%)
Reumatic disease	1 (4.5%)
Neurological disorders	1 (4.5%)
Atopic dermatitis	4 (18.2%)
Rhinoconjunctivitis	18 (81.8%)

**Table 2 children-09-00952-t002:** Spirometry values (expressed as percent of predicted values and as Z-score) of the recruited subjects.

	% Pred *	Z-Score *
FEV1	98.38 (94.38–104.39)	−0.14 (−0.48–0.37)
FVC	93.35 (89.56–103.56)	−0,51 (−0.89–0.3)
FEV1/FVC	105.34 (100.06–107.46)	0.9 (0.01–1.28)
FEF25-75	105.26 (92.35–117.33)	0.24 (0.35–0.77)

* Data are presented as median and interquartile range.

## Data Availability

The clinical documents of the current case report are available from the corresponding author on reasonable request.

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
