# Peer review of "Preliminary Evidence on Pulmonary Function after Asymptomatic and Mild COVID-19 in Children"

_children, 2022, doi:10.3390/children9070952_

Round 1

Reviewer 1 Report

The authors described the PF after SARS-CoV-2 infection in a pediatric cohort mostly presenting asymptomatic or mild disease, recruited at the Department of Women’s and Children’s Health (W&CHD) of Padua University Hospital (Italy).

The comments were added as follows,

The introduction was described clearly.

Methods: The inclusion criteria for this observational study were older than 6 years, right? Why did the researcher set this criterion? Or no child under 6 years has been included? Please mention the inclusion criteria.

 Results.

Line 86: What does “Tab 1” mean?

Please include the medicinal therapy and its time course (type of therapy, duration, etc) in Table 1 and the Results section. Is there any patient who received remdesivir or other anti-COVID-19 agents?

Please include the information about the history of vaccination (and its number).

.

Author Response

Padua, June, 9th, 2022

Dear Reviewer,

We thank you for your interest and the valuable comments on our manuscript.

We tried to answer the queries at our best. Enclosed you find our replies.

We are pleased to resubmit our work “Preliminary evidence on pulmonary function after asymptomatic and mild COVID-19 in children” after revision, according to your comments.

All authors have reviewed and agree with the new version of the manuscript. We hope you will find the revised manuscript acceptable for publication in the Children.

Sincerely,

Costanza Di Chiara, MD

Reply to Reviewer

We here summarize the amendments made to the manuscript.

Comments from Reviewer #1:

The authors described the PF after SARS-CoV-2 infection in a pediatric cohort mostly presenting asymptomatic or mild disease, recruited at the Department of Women’s and Children’s Health (W&CHD) of Padua University Hospital (Italy).

The comments were added as follows,

The introduction was described clearly.

Methods

Comment: The inclusion criteria for this observational study were older than 6 years, right? Why did the researcher set this criterion? Or no child under 6 years has been included? Please mention the inclusion criteria.

Reply: We thank the Reviewer for this relevant comment. We confirm that we included only subjects older than 6 years because they are able to perform spirometry following technical standards according to ERS guidelines, while preschoolers are not able to perform technically acceptable and repeatable spirometry. Therefore, in order to obtain reliable results, we decided to include only patients of suitable age to perform a good test. We added to the main text the inclusion criteria as suggested.

Change: Lines 57-58: Children younger than 6 years were not included because are not able to perform technically acceptable and repeatable spirometry.

Results

Comment: Line 86: What does “Tab 1” mean?

Please include the medicinal therapy and its time course (type of therapy, duration, etc) in Table 1 and the Results section. Is there any patient who received remdesivir or other anti-COVID-19 agents?

Please include the information about the history of vaccination (and its number).

Reply: We thank the Reviewer for this comment. Being totally affected by a mild or asymptomatic COVID-19, all enrolled subjects were treated as outpatients and did not receive any anti-SARS-CoV-2 therapies.

No patients also received any dose of any COVID-19 vaccine before being enrolled and tested. We specified this both in the results section of the text and in Table 1.

Change:

Lines86-87: “None children received any dose of COVID-19 vaccine neither before infection nor before pneumological evaluation.”

Lines 88-89: “None developed pneumonia, none received any anti-COVID-19 treatments.”

Reviewer 2 Report

In this prospective study the authors aimed to evaluate the long-term impact of mild/asymptomatic SARS-CoV-2 infection on the pulmonary function in children. The study was well designed and described and it brings new data to this topic. However, I have few concerns:

1. Methods point 2. " for asymptomatic cases with negative/undetermined nasal-pharyngeal swab" - if the child was asymptomatic and had a negative swab, there is no evidence of COVID-19 and these children should be excluded from the study.

2. Resuts: 66 children were evaluated, but only 61 underwent spirometry. Please explain why the remaining 5 children did not undergo the examination? Were these 66 children all the consecutive patients with COVID-19 in your Department? What were the indications for the spirometry? Please discuss.

3. Did the patients receive any treatment due to COVID-19, e.g. steroids?

4. Please explain all the abbreviations (even obvious) at the first time they are used (including SARS-CoV-2, COVID-19).

Author Response

Padua, June, 9th, 2022

Dear Reviewer,

We thank you for your interest and the valuable comments on our manuscript.

We tried to answer the queries at our best. Enclosed you find our replies.

We are pleased to resubmit our work “Preliminary evidence on pulmonary function after asymptomatic and mild COVID-19 in children” after revision, according to your comments.

All authors have reviewed and agree with the new version of the manuscript. We hope you will find the revised manuscript acceptable for publication in the Children.

Sincerely,

Costanza Di Chiara, MD

Reply to Reviewer

We here summarize the amendments made to the manuscript.

Comments from Reviewer #2:

In this prospective study the authors aimed to evaluate the long-term impact of mild/asymptomatic SARS-CoV-2 infection on the pulmonary function in children. The study was well designed and described and it brings new data to this topic. However, I have few concerns:

Methods

Comment:

point 2. "for asymptomatic cases with negative/undetermined nasal-pharyngeal swab" - if the child was asymptomatic and had a negative swab, there is no evidence of COVID-19, and these children should be excluded from the study.

Reply: We thank the Reviewer for this comment. At our COVID-19 family cluster follow-up Clinic, we enroll family clusters of COVID-19, including children/older siblings and their parents, sent to our evaluation from Family Pediatricians. At the first evaluation, we performed a clinical and serological evaluation to all family members. We collect clinical data on infections (data of positive nasopharyngeal swab, data of symptoms onset and symptoms duration, and data of close contact) and past medical history/vaccination, and we perform a blood sample collection to determine the antibodies response by chemiluminescence assay. Knowing features from all family members, we are able to estimate the data of infection onset also for those patients with no virological confirmation of COVID-19 but with a positive serological test after a close household contact. In those with only serologically confirmed COVID-19 and with negative/undetermined nasal-pharyngeal swab, by the family outbreak temporal sequence, coinciding with the date of symptoms onset in the family cluster.

We better specified in the method section as follow.

Change: Lines 58-63: “At enrolment, a pediatrician and/or an Infectious Diseases specialist collected data on demographic parameters, past medical history, clinical features (including date of symptoms onset, date of close household contacts, date of the positive nasopharyngeal swab), and vaccinal status (including the SARS-CoV-2 vaccine). A blood sample was collected from all cases for diagnosis of previous SARS-CoV-2 infection through the evaluation of the immunological response to SARS-CoV-2.”

Line 67: “for asymptomatic cases with negative/undetermined nasal-pharyngeal swab, the date was established within the family outbreak, coinciding with the date of symptoms onset in the family cluster.”

Results

Comment: 66 children were evaluated, but only 61 underwent spirometry. Please explain why the remaining 5 children did not undergo the examination? Were these 66 children all the consecutive patients with COVID-19 in your Department? What were the indications for the spirometry? Please discuss.

Reply: As we specified in the method section, we enrolled in this “pneumological function study” all children aged 6-18 years that were consecutively evaluated at our clinic during the study period, starting from August 2021. Children with a pre-existing chronic respiratory disease were excluded from the analysis, in order to evaluate the real impact of the SARS-CoV-2 virus on the pulmonary function without bias due to the presence of an underlying respiratory disease.

In our cohort, 5 children were excluded from the analysis because of a previous diagnosis of asthma (as reported in line 93 of the main text).

Change: None.

Comment: Did the patients receive any treatment due to COVID-19, e.g. steroids?

Reply: We thank the Reviewer for this comment. Being totally affected by a mild or asymptomatic COVID-19, all enrolled subjects were treated as outpatients and did not receive any anti-SARS-CoV-2 therapies. We specified this both in the result section of the text and in Table 1.

Change: Lines 88-89: “None developed pneumonia, or received any anti-COVID-19 treatments.”

Comment: Please explain all the abbreviations (even obvious) at the first time they are used (including SARS-CoV-2, COVID-19).

Reply: We thank the Reviewer for his/her comment. We explained all the abbreviations as suggested.

Change:

Line 23, abstract: Coronavirus disease 19 (COVID-19);

Lines 27,28, abstract: Severe Acute Respiratory Syndrome Coronavirus 2 (SARS-CoV-2);

Line 71, method: World Health Organization (WHO)

Round 2

Reviewer 1 Report

No additional revision will be required.